# Examination of unique volatile organic compound signatures in nasopharyngeal test swab viral transport media using an electronic nose

Barbara Swanson[1]*, Jessica Bishop-Royse[1], Ali Keshavarzian[2], Robert Balk[2], Abhinav Bhushan[3], James Moy[2], Alan Landay[4], Wrenetha Julion[1], Jhalak Mehta[5], Maryan Arrieta[1], Dylan Behun[2], Michael Bowlen[6], Minnie Kang[2]

1 College of Nursing, Rush University, Chicago, United States of America, 2 College of Medicine, Rush University, Chicago, United States of America, 3 Biomedical Engineering, Illinois Institute of Technology, Chicago, United States of America, 4 Research Services, The University of Texas Medical Branch at Galveston School of Medicine, United States of America, 5 College of Medicine, Baylor University, United States of America, 6 Chicago Medical School, Rosalind Franklin University of Medicine and Science, North Chicago, United States of America

* barbara_a_swanson@rush.edu

## Abstract

Rapid point of care tests for respiratory infections are associated with high rates of false negative results which can drive empiric, and potentially inappropriate, antibiotic use. Because infectious pathogens alter VOC composition, unique VOC signatures in biospecimens hold the potential to discriminate bacterial and viral infections from uninfected controls. One approach for rapid identification of respiratory pathogens is the electronic nose (e-nose), a sensor device that uses artificial intelligence to recognize disease-specific patterns in VOC profiles of gaseous mixtures. In this preclinical proof of concept study, we tested the validity of an e-nose to discriminate PCR-confirmed cases of infection with three viral pathogens (SARS-CoV-2, RSV, influenza A) from uninfected controls using nasopharyngeal test swab media. Using exploratory factor analysis, the e-nose discriminated both influenza A and SAR-CoV-2 from uninfected controls. To assess sensitivity and specificity, we applied factor analysis-based threshold values and obtained high levels of sensitivity (96.30%) and specificity (90.62%) for influenza A and more modest levels for SARS-CoV-2 (sensitivity=75%, specificity=68.57%). We did not apply threshold values to RSV samples because the e-nose sensors showed low discriminatory power for that pathogen. Our findings support proof of concept of the validity of the e-nose to discriminate common viral respiratory pathogens. Our use of binary thresholds for influenza A, which are easily adapted to point-of-care settings, yielded superior sensitivity results and comparable specificity results when compared to rapid tests. We recommend that future studies apply our analytic approach to samples of human breath to determine if these findings can be replicated or improved.

**Data availability statement:** All relevant data are within the paper and its Supporting Information files.

**Funding:** Funding from Center for Emerging Infectious Diseases, Rush University Medical Center, Chicago, IL.

**Competing interests:** The authors have declared that no competing interests exist.

## Introduction

Inappropriate antibiotic therapy for viral respiratory infection in the absence of concurrent bacterial infection is a leading cause of antimicrobial resistance [1]. Rates of antibiotic use for viral infections have been reported to range from 14% to 83% [2,3]. Moreover, the COVID-19 pandemic exacerbated antibiotic overprescription. Between March, 2020 and August, 2021, approximately half of COVID-19 inpatients received ceftriaxone suggesting limitations discriminating SARS-CoV-2 from community-acquired pneumonia [4]. While rapid point of care tests with acceptable sensitivity and specificity are now available for SARS-CoV-2, rapid tests for respiratory syncytial virus (RSV) and influenza are associated with high rates of false negative results [5,6]. This lack of sensitivity can drive empiric, and potentially inappropriate, antibiotic use.

One method for rapidly screening common respiratory pathogens is analysis of volatile organic compounds (VOC), gases that are endogenously formed as metabolic byproducts [7]. Infectious pathogens alter VOC composition, thus unique VOC signatures in exhaled breath and supernatants of infected cell lines have discriminated bacterial and viral infections from uninfected controls [8–10]; however, to our knowledge, there have been no studies comparing the unique VOC signatures among viral respiratory pathogen infections. Determining unique viral pathogen VOC signatures holds the potential to promote accelerated administration of appropriate antiviral therapies and avert antibiotic misuse.

One innovative approach for rapid identification of respiratory pathogens is the electronic nose (e-nose). The e-nose is a sensor device that uses artificial intelligence to recognize disease-specific patterns in VOC profiles of gaseous mixtures. The e-nose contains an array of carbon-based sensors coated with a non-conducting polymer surface. Individual VOC adsorb onto multiple sensors and induce swelling of the polymer coat and a change in the sensors' electrical resistance [11] (see Fig 1). The VOC profile creates a unique electrical resistance response, known as a smell-print, that can be analyzed using algorithms that recognize disease-specific patterns [12]. A growing body of evidence has supported the validity of e-nose-generated breath smellprints to discriminate persons with various diseases from healthy controls, including lung infections [12–14].

To our knowledge, there have been no studies to investigate the discriminant validity of viral transport media (VTM) smellprints collected from persons who present with respiratory symptoms. VOC signatures of remnant VTM specimens can be quickly analyzed at point of care using an e-nose, thus analyses are warranted to determine if they yield superior sensitivity and specificity results than commercially available rapid point of care tests. In this preclinical proof of concept study, we investigated the presence of unique VOC signatures in nasopharyngeal test swab media collected from persons with PCR-confirmed viral infections and uninfected controls using an e-nose. We hypothesized that the e-nose would discriminate VOC signatures in swab media from persons with infection with SARS-CoV-2, RSV, and influenza A, as well as uninfected controls.

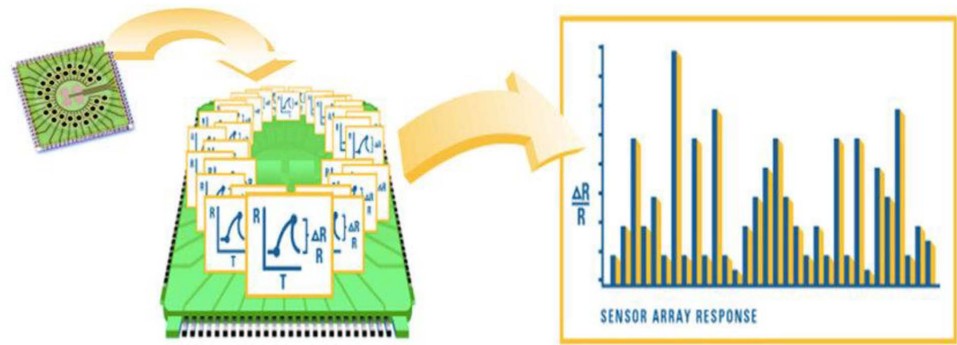

The electronic nose contains an array of polymer sensors that change resistivity based on adsorption of VOC. Because each sensor is unique, the analysis yields unique resistance changes that comprise the response.

Reproduced with Creative Commons license from: Dragonieri S, Quaranta VN, Portacci A, Ahroud M, Di Marco M, Ranieri T, Carpagnano GE. Effect of food intake on exhaled volatile organic compounds profile analyzed by an electronic nose. *Molecules*. 2023 Jul 30;28(15):5755.

**Fig 1. Mechanism of Electronic Nose.**

## Materials and methods

### Participants and materials

This study was submitted to the Institutional Review Board of Rush University Medical Center (RUMC) who determined that it was non-human subjects research and exempted from review. Nasopharyngeal swabs were collected from adults and children presenting to RUMC clinics with symptoms of upper respiratory infection between November, 2022 and February, 2023. All swabs were placed in viral transport media (VTM) and analyzed in a BSL-2 laboratory in the Rush Division of Clinical Microbiology using a molecular multiplex assay (MMPA). The MMPA can simultaneously confirm the presence of SARS-CoV-2, RSV, and influenza A. All VTM specimens and MMPA results were de-identified. VTM specimens that were obtained from dually infected individuals were excluded. Remnant VTM specimens of approximately 1.8 milliliters were transferred to conical tubes and stored in a -70$^\circ$C freezer until e-nose analyses were performed. Immediately prior to e-nose analyses, frozen VTM specimens were thawed for 20 minutes in a 4$^\circ$C refrigerator and analyzed under a laminar flow hood. One ml of each specimen was transferred to a 40 ml autosampler vial with a rubber stopper screw top.

### E-nose analyses

All analyses of specimens were performed using the Cyranose 320 e-nose (Sensigent, Baldwin Park, CA). The Cyranose 320 contains an array of 32 nanocomposite sensors that change resistivity based on adsorption of VOC. Because each sensor is unique, the analysis yields 32 unique resistance changes that are calculated as $(R_{max} - R_0)/R_0$, where $R_{max}$ is the sensor's maximum resistance and $R_0$ is the baseline resistance. The Cyranose 320 has shown high levels of reproducibility for biological specimens including stool (five replicate measurements of three separate stool specimens showed mean ICC = 0.997 for each specimen) [15] and breath (within day ICC range = 0.75 to 0.84; 7-day between-day ICC range = 0.57 to 0.76) [16].

 In accordance with the manufacturer's instructions, prior to specimen analyses the Cyranose 320 sampled and purged room air for six minutes to establish a baseline exposure to the ambient environment. Next, a needle was inserted to vent

the vial, thus equilibrating pressure and promoting reproducibility, and the snout of the e-nose punctured the septum to sample the headspace. Each specimen was analyzed once following a procedure of 10 seconds of sampling ambient air, 10 seconds of sampling the specimen headspace, and 35 seconds of purging.

### Data analysis

All data analyses were performed using Stata v. 18 (College Station, TX). Because the sampling of each specimen's headspace for 10 seconds yielded multiple values for each sensor, we calculated the mean score for each individual sensor for each specimen's readings, after excluding the first reading per the guidance of the e-nose manufacturer. We then performed exploratory factor analysis (EFA) with oblique rotation using the mean values for each of the 32 sensors. We utilized this approach over others, such as random forests (RF) and support vector machine (SVM), due to our limited sample size (N = 119). Other machine-learning approaches, including RF and SVM, are computationally intense and require a large event to variable ratio [16]. For the present study, this would have required a sample size of 330. Also, because we did not have well-defined, theoretically-based factor structures for confirmation, we chose to utilize EFA over confirmatory factor analysis.

### Results

We obtained VTM specimens from 119 individuals with the following MMPA results: SARS-CoV-2 (n = 29; 24.37%), RSV (n = 31; 26.05%), influenza A (n = 29; 24.37%), and negative for all tested pathogens (n = 30; 25.21%).

Exploratory factor analysis on all 32 sensors yielded a three-factor solution that accounted for 97.4% of the variation. Parallel analysis, which is the gold standard for dimension reduction and measure validation, [17] suggested discarding Factor 3 and retaining Factors 1 and 2; however, because Factor 2 consisted of only one sensor, it was discarded. Thus all subsequent analyses were performed on Factor 1 data.

The rotated factor loadings were relatively consistent and high for Factor 1, although 10 sensors showed factor loadings below 0.90. While it has been suggested that variables with factor loadings above 0.50 should be retained for sample sizes comparable to ours (N = 119), [18] we selected a higher threshold (0.90) because we examined sensor data, not variables. Using these data, which also allowed for strong correlations among sensors, we applied this higher threshold for factor loadings, rather than the commonly accepted 0.50. Using this threshold, ten sensors were eliminated from the analyses, leaving 22 sensors to comprise Factor 1.

Bartlett's test of sphericity was significant ($X^2$ = 12209.606; p ≤ 0.0001), suggesting that Factor 1 sensor values were not intercorrelated, and Kaiser-Meyer-Olkin Measure of sampling adequacy was 0.974, indicating that our sample size was adequate to support the analyses. Mean scores for each specimen type are shown in Table 1. There were statistically significant differences in average Factor 1 scores by specimen type (F = 21.27, p ≤ 0.001). Average Factor 1 scores were largest for influenza A (0.0786), followed by SARS-CoV-2 (0.0698) and RSV (0.0638). The mean Factor 1 score for uninfected control specimens was (0.0644).

Next, we examined the sensitivity and specificity of Factor 1 for discriminating between infected and uninfected specimens using receiver operating curves (ROC). As shown in Fig 2, Factor 1 values had the least power for discriminating

**Table 1. Specimen Types and Mean Scores on Factor 1.**

| Type | N (%) | Factor 1 Mean Score (SD) | ANOVA |
|---|---|---|---|
| SARS-CoV-2 | 29 (24.37%) | 0.0698 (0.001) | F = 21.27, p ≤ 0.001 |
| RSV | 31 (26.05%) | 0.0638 (0.002) | |
| Influenza A | 29 (24.37%) | 0.0786 (0.001) | |
| Uninfected control | 30 (25.21%) | 0.0644 (0.001) | |

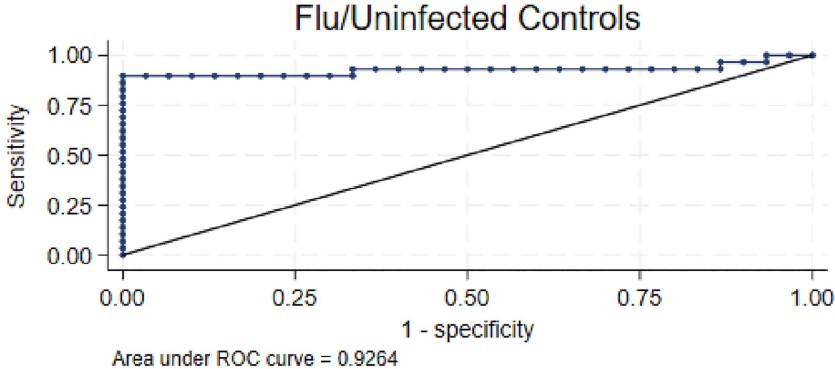

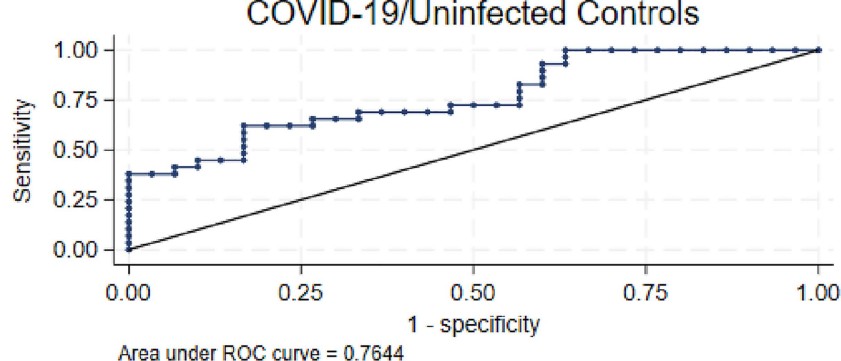

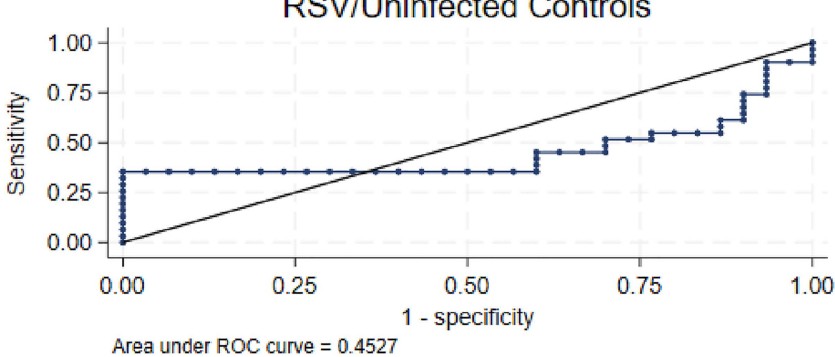

**Fig 2. ROC Comparing Pathogen Type to Uninfected Controls.**

between RSV and uninfected specimens (ROC = 0.4527) and were acceptable for discriminating between SARS-CoV-2 and uninfected specimens (ROC = 0.7644). Notably, ROC values were highest for discriminating between influenza A and uninfected specimens (ROC = 0.9264).

Fig 3 shows ROC curves for Factor 1 comparisons between specimens that were or were not positive for a given pathogen and were or were not uninfected controls. The best iteration of these analyses was for Factor 1 scores of influenza A specimens compared to non-influenza A specimens. The associated area under the curve was the largest of any of the comparisons (0.898). Each of the other comparisons, RSV to non-RSV, SARS-CoV-2 to non-SARS CoV-2, and uninfected control to non-uninfected control produced ROC curves with unacceptably low scores.

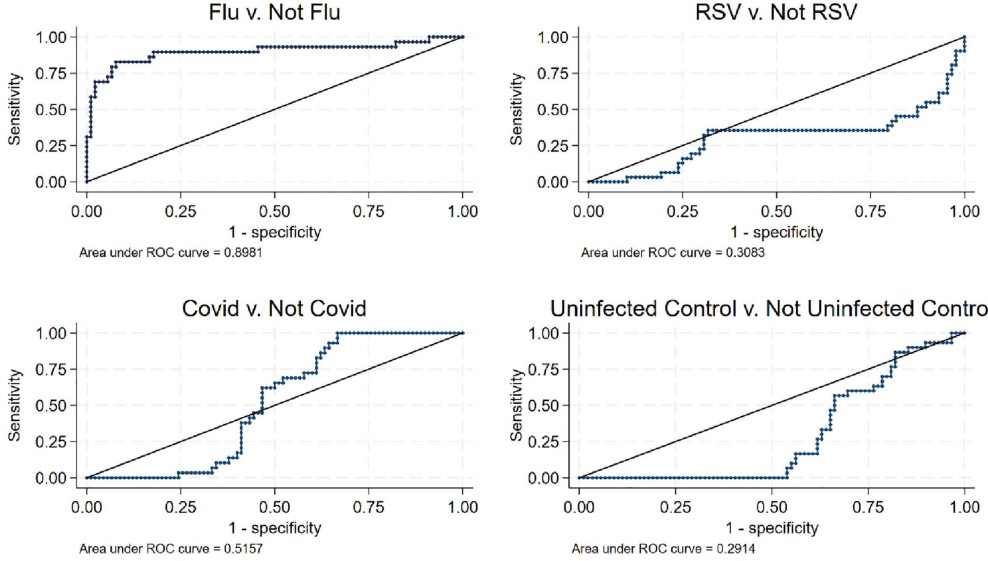

**Fig 3. ROC Analysis to Compare Specimen Types.**

Detailed output from the ROC analyses suggested values at which Factor 1 scores could be used to discriminate between uninfected specimens and both influenza A and SARS-CoV-2 specimens. To further test this proof of concept, we calculated two binary variables and performed two additional ROC analyses. The first analysis utilized a binary variable for influenza A that was constructed using the threshold of 0.0714. This value was determined from the initial ROC analysis reported above and corresponded to the level at which influenza A infected specimens were largely predicted correctly. Factor 1 scores above 0.0714 were assigned as influenza A probable. When this designation was compared to the actual influenza A specimens, we found that 26 of the 27 specimens with scores greater than 0.0714 were influenza A, while 29 of the 32 specimens with values below 0.0714 were uninfected. Table 2 shows sensitivity of 96.30% and specificity of 90.62%, with an area under the curve of 0.93.

These analyses were repeated with the lower Factor 1 scores indicated for SARS-CoV-2 specimens. A binary variable was calculated using the threshold of 0.0693 to determine whether these procedures could reasonably predict SARS-CoV-2. With this measure, higher values were considered SARS-CoV-2 positive probable, lower values were assigned SARS-CoV-2 negative probable. Based on this lower threshold, 18 of 24 specimens were correctly predicted to be

**Table 2. Comparisons of Specimen Prediction to Actual Specimen Type.**

| Specimen Type | No Flu Probable (N/%) | Yes Flu Probable (N/%) | Total | Discriminant Validity |
|---|---|---|---|---|
| Uninfected control | 29 (96.66%) | 1 (3.34) | 30 | Sensitivity = 96.30% |
| Influenza A | 3 (10.34%) | 26 (89.65%) | 29 | Specificity = 90.62% |
| Total | 32 | 27 | 59 | PPV = 89.66% NPV = 96.67% ROC Area = 0.93 |
| **Specimen Type** | **No COVID-19 Probable (N/%)** | **Yes COVID-19 Probable (N/%)** | **Total** | **Discriminant Validity** |
| Uninfected control | 24 (80.00%) | 6 (20.00%) | 30 | Sensitivity = 75.00% |
| COVID-19 | 11 (37.93%) | 18 (62.07%) | 29 | Specificity = 68.57% |
| Total | 35 | 24 | 59 | PPV = 62.07% NPV = 80.00% ROC Area = 0.71 |

SARS-CoV-2. Additionally, we found that 24 of 35 specimens were correctly predicted to be uninfected. ROC analysis showed sensitivity of 75% and specificity of 68.57%, with ROC area of 0.71.

## Discussion

We showed that a commercially available e-nose could discriminate unique VOC signatures in nasopharyngeal swab VTM. Specifically, using data from 22 sensors that were identified using EFA, we were able to discriminate both influenza A and SAR-CoV-2 from uninfected controls, but we did not find acceptable power to discriminate RSV from uninfected controls. These findings are consistent with a growing body of e-nose studies that have found unique VOC signatures in breath and nasal specimens obtained from persons with confirmed viral or bacterial respiratory infection [8,19–23]

We extended the body of e-nose literature by applying factor analysis-based threshold values to assess sensitivity and specificity to discriminate viral infections from uninfected controls. Our model showed high levels of sensitivity (96.30%) and specificity (90.62%) for influenza A and more modest levels for SARS-CoV-2 (sensitivity = 75%, specificity = 68.57%). These findings are significant because they were derived from an analytic approach designed to promote the use of e-nose technology for point of care testing. Commonly used approaches to analyze e-nose data include principal component analysis and neural networks, neither of which is feasible for point of care testing and interpretation. In contrast, binary threshold values that discriminate cases from controls scores are easily interpreted at point of care and hold the potential to avert inappropriate antibiotic use pending return of confirmatory test results. Sensitivity and specificity analyses, as well as the ROC area, suggest that these procedures are more robust for identifying influenza A specimens than for SARS-CoV-2 specimens.

We did not apply threshold values to RSV specimens because the Factor 1 sensors showed low discriminatory power for that pathogen. It is not clear why the discriminatory power for RSV was lower than influenza A and SARS-CoV-2. All three pathogens are RNA viruses with low fidelity polymerases [24,25], suggesting that intravariant differences in VOC signatures are not a likely explanation for this finding. In support of this conclusion, we found the highest discriminatory power for influenza A, a virus whose genomic mutation rate has been shown to be 23.9 fold higher than SARS-CoV-2 [26]. It is also possible that differences in the induced immune responses explained the low discriminatory power. Compared to influenza A, RSV induces a more dysregulated immune response characterized by weak T cell responses and higher risk of strain-specific reinfection [27]. Additionally, RSV replication kinetics are slower than influenza A potentially affecting the timing of peak VOC expression and discriminatory power [28].

We chose not to separate our specimens into training and independent validation datasets. In this proof of concept study, our priority was to explore whether the respiratory pathogens showed unique VOC signatures. Our findings support follow-up studies to test whether the obtained algorithm can discriminate pathogen-associated VOC signatures in an independent dataset.

One limitation was the use of VTM, rather than breath, specimens. We chose to analyze VTM specimens because (a) there is no consensus on a standardized protocol for collecting breath for VOC signatures, thus the validity of the results may be compromised by the failure to control poorly characterized/unknown extraneous sources of variance, and (b) nasopharyngeal swabs, rather than breath specimens, are obtained at point of care for rapid respiratory pathogen identification. Consequently, we chose to concurrently analyze VTM specimens with an alternative technology to determine if it produced superior sensitivity and specificity results.

A second limitation was the absence of participant demographic and clinical information that may have informed interpretation of the findings. Smoking status, age, medications, and co-morbidities, such as diabetes mellitus and asthma, have been associated with unique VOC signatures [29–32]. Accordingly, we cannot rule out that heterogeneity in these parameters across the viral infection groups limited our ability to discriminate RSV from the other pathogens or to achieve higher discriminatory power for SARS-CoV-2.

A third limitation is the possibility that our use of high thresholds for eigenvalues resulted in an overfitted model. Our interest was in identifying the sensors most likely to differentiate among different pathogens encountered in a clinical setting. Initial EFA results returned eigenvalues that were high for all the sensors, and our choice to retain only those that were above 0.90 may have resulted in an overfitted and less generalizable model. Methods to assess EFA overfitting include checking factor stability in split samples, examining model communalities, and evaluating differences relative to alternative extraction methods. As described below, we performed these analyses and found mixed results.

We split the sample into two random subsets and repeated the EFA procedures on each subset, as stable factor loadings across subsets suggest less risk of overfitting [33]. We found that the rotated factor loadings for the subsets (n = 59 and n = 60) were similar to those for the full sample with two exceptions. For subset 1 (n = 60), three sensors that had been excluded in the full sample due to loadings below.90 had sensor values above 0.90. For subset 2 (n = 59), one sensor that had been excluded for loadings below 0.90 had factor loadings above 0.90. It should be noted that in each of these cases, the factor loadings in the subset analysis were less than 0.91.

Our examination of model communalities returned results consistent with an overfitted model. When the 22 sensors retained for inclusion in the instrument were subjected to additional factor analysis, $h^2$ or communalities values were higher than 0.80 which may indicate overfitting [34].

To assess differences compared to an alternative extraction method, we replicated our analyses using principal components factors and maximum likelihood extraction. We found nearly identical results which argued against an overfitted model.

Given these findings, we are unable to draw conclusions about the overfitting of our model. The most substantial concern with overfitted results is a lack of generalizability. Given the exploratory nature of this study, we take a somewhat liberal view on whether our use of a high threshold for eigenvalues resulted in less generalizable findings. We concede that it is possible that our approach does not generalize well to other populations or contexts. More research in this area will help determine if this approach using e-nose data and methods is viable.

In conclusion, our findings support proof of concept of the validity of the e-nose to discriminate common viral respiratory pathogens. Moreover, our use of binary thresholds for influenza A, which are easily adapted to point-of-care settings, yielded superior sensitivity results (96.3%) and comparable specificity results (90.62%) when compared to rapid tests (sensitivity range = 50–70%; specificity range = 95–99%) [6]. We recommend that future studies apply our analytic approach on specimens of human breath to determine if these findings can be replicated or improved.

## Supporting information

**S1. Collection 1.**
(CSV)

**S2. Collection 2.**
(CSV)

**S3. Collection 3.**
(CSV)

**S4. Collection 4.**
(CSV)

**S5. Collection 5.**
(CSV)

**S6. Collection 6.**
(CSV)

**S7. Collection 7.**

(CSV)

**S8. Collection 8.**

(CSV)

**S9. Collection 9.**

(CSV)

**S10. Collection 10.**

(CSV)

## Author contributions

**Conceptualization:** Barbara Swanson, Ali Keshavarzian, Robert Balk, Abhinav Bhushan, Alan Landay, Wrenetha Julion.

**Formal analysis:** Jessica Bishop-Royse.

**Funding acquisition:** Barbara Swanson.

**Investigation:** Dylan Behun, Michael Bowlen.

**Methodology:** Barbara Swanson, Jessica Bishop-Royse.

**Supervision:** Barbara Swanson.

**Writing – original draft:** Barbara Swanson, Jessica Bishop-Royse, Ali Keshavarzian, Robert Balk, Abhinav Bhushan, James Moy, Wrenetha Julion, Jhalak Mehta, Maryan Arrieta, Dylan Behun, Michael Bowlen, Minnie Kang.

**Writing – review & editing:** Barbara Swanson, Jessica Bishop-Royse, Ali Keshavarzian, Robert Balk, Abhinav Bhushan, James Moy, Alan Landay, Wrenetha Julion, Jhalak Mehta, Maryan Arrieta, Dylan Behun, Michael Bowlen, Minnie Kang.

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
