## [Decision Letter · Decision Letter 0]

4 Mar 2025

PONE-D-24-56295Examination of Unique Volatile Organic Compound Signatures in Nasopharyngeal Test Swab Viral Transport Media Using an Electronic NosePLOS ONE

Dear Dr. swanson,

Thank you for submitting your manuscript to PLOS ONE. After careful consideration, we feel that it has merit but does not fully meet PLOS ONE’s publication criteria as it currently stands. Therefore, we invite you to submit a revised version of the manuscript that addresses the points raised during the review process.

We look forward to receiving your revised manuscript.

Kind regards,

Rajeev Singh

Academic Editor

PLOS ONE

Journal Requirements:

2. Thank you for stating the following financial disclosure: [Funding from Center for Emerging Infectious Diseases, Rush University Medical Center, Chicago, IL].

Reviewers' comments:

Reviewer's Responses to Questions

**Comments to the Author**

1. Is the manuscript technically sound, and do the data support the conclusions?

Reviewer #1: Yes

Reviewer #2: Partly

2. Has the statistical analysis been performed appropriately and rigorously? 

Reviewer #1: Yes

Reviewer #2: Yes

3. Have the authors made all data underlying the findings in their manuscript fully available?

Reviewer #1: Yes

Reviewer #2: No

4. Is the manuscript presented in an intelligible fashion and written in standard English?

Reviewer #1: Yes

Reviewer #2: Yes

5. Review Comments to the Author

Reviewer #1: 1) The novelty of the work should be clearly indicated in the manuscript in introduction section.

2) what are the units for x and y axis in the figures? They should be added.

3) Figure qualities should be improved.

Reviewer #2: I read this paper with great interest. Authors showed that a Cyranose could detect VOCprofiles in VTM from patients with respiratory viral infections. It was ok for influenza A and COVID vs controls, but bad for RSV.

The study is interesting and quite well-written but it has several major concerns:

- The rationale for selecting EFA over PCA or other machine learning-based approaches (e.g., SVM or random forests) is not well justified. Please comment on that.

-The threshold values for Factor 1based binary classification need further validation. Were these thresholds derived from an independent dataset or via cross-validation? If not, the risk of overfitting must be discussed.

-A power calculation should be provided to justify whether your sample is sufficient for robust statistical conclusions.

- The study lacks external validation with a separate dataset or cohort. Without validation, it is difficult to generalize the findings to real-world applications.

- The headspace analysis of VTM may not directly translate to patient breath samples, limiting clinical applicability. Please comment on that.

- The Cyranose failed to differentiate RSV from uninfected samples. This limitation must be explored further (e.g. was this due to low VOC concentrations, differences in viral replication/metabolism, or sensor limitations?) What alternative strategies can be employed to enhance sensitivity for RSV?

- An eye-catching figure illustrating the mechanism of e-nose detection of viral infections is needed.

- The discussion is really too short for an original article, unless you choose for a brief report.

6. PLOS authors have the option to publish the peer review history of their article (what does this mean?). If published, this will include your full peer review and any attached files.

Reviewer #1: **Yes: **Tugba Ozer

Reviewer #2: No

---

## [Author Response · Author response to Decision Letter 1]

14 Aug 2025

We appreciate the reviewers’ thoughtful comments and have responded as described below.

REVIEWER #1

• The novelty of the work should be clearly indicated in the manuscript in the introduction section

o We have added a statement on p. 3 (starting on line 9) to indicate the novelty of our study.

• What are the units for x and y axis in the figures? They should be added

o The x axis represents the false positive rate, while the y axis represents the true positive rate. We have labeled these axes as specificity (x axis) and sensitivity (y axis).

• Figure qualities should be improved

o We have improved the resolution of the figures for readability

REVIEWER #2

• The rationale for selecting EFA over PCA or other machine learning-based approaches (e.g., SVM or random forests) is not well justified. Please comment on that.

o We have added a rationale for selecting EFA on p. 5.

• The threshold values for Factor 1-based binary classification need further validation. Were these thresholds derived from an independent dataset or via cross-validation? If not, the risk of overfitting must be discussed.

o We have expanded our discussion of the binary classification system that we used and the potential for overfitting (pp. 10-11).

• A power calculation should be provided to justify whether your sample is sufficient for robust statistical conclusions.

o The analyses were selected to align with our proof-of-concept design. We were interested in exploratory factor analysis results because they provide qualitatively different information than what can be provided using hypothesis testing. We utilized a convenience sample without considering statistical power because EFA procedures are not based on a hypothesis testing model that generates a single p-value. Consequently, it is not possible to calculate the true effect size that would be required to perform a post-hoc power analysis. Instead, we performed a Kaiser-Myer-Olkin test to determine our sample size adequacy for EFA (p. 6).

We also did not perform a power calculation for the ANOVA results as it would not have informed our methods. We specified an alpha of 0.05 which gave us a 5% chance of incorrectly rejecting the null hypothesis (that there are no differences in mean Factor 1 scores by sample type). Power calculations are most useful when they are specified a priori, when estimations of effect size and error are known, as they can reduce the likelihood of underpowered studies. Post-hoc power calculations to contextualize or interpret results is inappropriate and often produces misleading results, as noted by Heckmen et al (2022) and Zhang et al (2019).

• The study lacks external validation with a separate dataset or cohort. Without validation, it is difficult to generalize the findings to real-world applications.

o We made the decision not to separate our samples into training and validation sets because, consistent with our proof of concept design, we wanted to use all samples to explore whether the device could discriminate pathogen-specific VOC signatures. Since our findings suggested that the device could discriminate two pathogens from uninfected controls, we believe that the next step is to train the device to recognize those pathogens’ signatures and to test the obtained algorithm on an independent validation dataset. We have added this to the discussion section (p. 9, second paragraph).

• The headspace analysis of VTM may not directly translate to patient breath samples, limiting clinical applicability. Please comment on that.

o We addressed this as a limitation (p. 10, first paragraph).

• The Cyranose failed to differentiate RSV from uninfected samples. This limitation must be explored further (e.g., was this due to low VOC concentrations, differences in viral replication/metabolism, or sensor limitations?) What alternative strategies can be employed to enhance sensitivity for RSV?

o We have provided potential explanations for the inability to discriminate RSV from uninfected samples (p. 9, first paragraph).

• An eye-catching figure illustrating the mechanism of e-nose detection of viral infections is needed.

o We have added a figure that depicts the e-nose mechanism.

• The discussion is really too short for an original article, unless you choose for a brief report.

o We have expanded our discussion section to make it more appropriate for an original article.

---

## [Editor Report · Decision Letter 1]

31 Aug 2025

Examination of Unique Volatile Organic Compound Signatures in Nasopharyngeal Test Swab Viral Transport Media Using an Electronic Nose

PONE-D-24-56295R1

Dear Dr. swanson,

We’re pleased to inform you that your manuscript has been judged scientifically suitable for publication and will be formally accepted for publication once it meets all outstanding technical requirements.

Kind regards,

Rajeev Singh

Academic Editor

PLOS ONE
---

## [Editor Report · Acceptance letter]

PONE-D-24-56295R1

PLOS ONE

Dear Dr. swanson,

I'm pleased to inform you that your manuscript has been deemed suitable for publication in PLOS ONE. Congratulations! Your manuscript is now being handed over to our production team.

Kind regards,

on behalf of

Dr. Rajeev Singh

Academic Editor

PLOS ONE